# Dental microwear of a basal ankylosaurine dinosaur, *Jinyunpelta* and its implication on evolution of chewing mechanism in ankylosaurs

Tai Kubo[1]*, Wenjie Zheng[2], Mugino O. Kubo[3], Xingsheng Jin[2]

**1** The University Museum, The University of Tokyo, Tokyo, Japan, **2** Zhejiang Museum of Natural History, Hangzhou, Zhejiang, People's Republic of China, **3** Department of Natural Environmental Studies, Graduate School of Frontier Sciences, The University of Tokyo, Kashiwa, Chiba, Japan

* taikubo@um.u-tokyo.ac.jp

**Data Availability Statement:** All relevant data are within the paper and its Supporting information files.

## Abstract

*Jinyunpelta sinensis* is a basal ankylosaurine dinosaur excavated from the mid Cretaceous Liangtoutang Formation of Jinyun County, Zhejiang Province, China. In the present study, its dental microwear was observed using a confocal laser microscope. *Jinyunpelta* had steep wear facets that covered most of buccal surfaces of posterior dentary teeth. Observation of dental microwear on the wear facet revealed that scratch orientation varied according to its location within the wear facet: vertically (i.e. apicobasally) oriented scratches were dominant in the upper half of the wear facet, and horizontally (i.e. mesiolaterally) oriented ones were in the bottom of the facet. These findings indicated that *Jinyunpelta* adopted precise tooth occlusion and biphasal jaw movement (orthal closure and palinal lower jaw movement). The biphasal jaw movement was widely observed among nodosaurids, among ankylosaurids, it was previously only known from the Late Cretaceous North American taxa, and not known among Asian ankylosaurids. The finding of biphasal jaw movement in *Jinyunpelta* showed sophisticate feeding adaptations emerged among ankylosaurids much earlier (during Albian or Cenomanian) than previously thought (during Campanian). The Evolution of the biphasal jaw mechanism that contemporaneously occurred among two lineages of ankylosaurs, ankylosaurids and nodosaurids, showed high evolutionary plasticity of ankylosaur jaw mechanics.

## Introduction

Ankylosaurs were herbivorous dinosaurs that emerged in the Middle Jurassic and prospered until the end of Cretaceous [1, 2]. During the Late Cretaceous, compared to other contemporaneous megaherbivores, such as ceratopsians and hadrosaurs, ankylosaurs likely fed on less fibrous plants growing in lower layers of paleovegetation [3, 4]. Ankylosaurs were assumed to feed mainly on herbaceous ferns [4], which were dominant in stomach contents of an Early Cretaceous nodosaurid ankylosaur, *Borealopelta*, implying that this animal fed on ferns

**Funding:** TK was funded by JSPS KAKENHI grant number 19J40003. WZ and XJ were funded by Chinese Natural Science Foundation (41602019). The funders had no role in study design, data collection and analysis, decision to publish, or preparation of the manuscript.

**Competing interests:** The authors have declared that no competing interests exist.

selectively [5]. Stomach contents of another Early Cretaceous basal ankylosaur, *Kunbarrasaurus*, also contained possible fern sporangia along with vascular tissue, angiosperm fruits, and small seeds [6, 7]. Between two main ankylosaurian families, i.e. Nodosauridae and Ankylosauridae, the former was assumed to have consumed more fibrous and tougher food than the latter, but dental microwear of Late Cretaceous North American taxa did not show distinction between these two families [3]. In addition to diet, jaw mechanics are revealed via ankylosaur dental microwear. Contrary to the expectation that ankylosaurs adopted simple orthal jaw movement due to their small and simple leaf-shaped teeth [8–10], dental microwear of derived Late Cretaceous taxa, such as an ankylosaurid *Euoplocephalus* [11] and nodosaurids, *Panoplosaurus* and *Hungarosaurus*, [3, 12] indicated biphasal jaw mechanism that adopted both orthal closure and palinal lower jaw movements.

It has been shown by analyses of skull morphometrics and dental wear that ankylosaurs evolved the biphasal jaw mechanism convergently in different ankylosaurian lineages [13]. According to Ősi et al. [12], basal ankylosaurs used orthal jaw movement, during which teeth rarely occluded. Precise tooth occlusion evolved convergently among basal nodosaurids and at the common ancestor of derived North American ankylosaurids. An Early Cretaceous nodosaurid *Sauropelta* might have adopted palinal jaw movements. This would also be the cases for Late Cretaceous nodosaurids, *Struthiosaurus* and *Panoplosaurus* as well as for a Late Cretaceous ankylosaurid *Ankylosaurus*. On the other hand, there were clear evidences of palinal jaw movements for Late Cretaceous nodosaurids *Edmontonia*, *Panoplosaurus*, *Hungarosaurus*, and a Late Cretaceous ankylosaurid *Euoplocephalus* [13]. Asian ankylosaurids, however, were thought to retain ancestral orthal jaw movement without tooth occlusion [13]. The evolutionary pattern of jaw mechanism in ankylosaurs is more complex than those of other herbivorous dinosaur taxa. For example, hadrosaurids exhibited a continuous evolutionary trend toward a more efficient masticatory system throughout the Cretaceous, such as an increase of functional teeth and the invention of new dental tissue types [14, 15]. The dental microwear of more ankylosaur specimens is needed to determine if the many convergences found by Ősi [13] are the true pattern of evolution or an artifact of sample size. Also, to deduce the factors that promoted the convergent evolution of sophisticated feeding adaptations in different ankylosaurian lineages, data from ankylosaur species that lived in various regions and periods were needed. In this study, we observed the dental microwear of *Jinyunpelta* to deduce its jaw movement and to reconstruct the evolution of the feeding mechanism in Asian and Cretaceous ankylosaurids.

## Materials and methods

*Jinyunpelta sinensis* is the most basal ankylosaurine and the oldest ankylosaur with a tail club, which was excavated by a joint team of Zhejiang Museum of Natural History (ZMNH), Jinyun Museum, and Fukui Prefectural Dinosaur Museum during 2013 from Albian–Cenomanian Liangtoutang Formation of Jinyun County, Zhejiang Province, China [16]. As the excavation was conducted by the provincial museum, Zhejiang Museum of Natural History, and the fossil site was in the Zhejiang province of China, no permits were required for the excavation. The purpose of the excavation was to rescue fossil specimens from the reclamation work that completely flatten the hill, which contained these fossils. During the excavation, five associated specimens and isolated materials of *Jinyunpelta* were found from the construction site. The subsequent description was based on two specimens (Holotype ZMNH M8960 and Paratype ZMNH M8963) as other specimens were still under preparation [16]. The holotype skull ZMNH M8960 is dorsoventrally compressed and its upper and lower teeth rows are tightly occluded, therefore existence or absence of wear facet(s) on teeth could not be determined.

Recently, it was clarified through preparation that another *Jinyunpelta* specimen ZMNH M8961 conserved the posterior part of dentaries and maxillae (Fig 1). The specimen is publicly deposited and accessible by others in a permanent repository as ZMNH is a provincial museum. Due to postmortem damages, most teeth were fallen out from teeth sockets and surfaces of many *in situ* teeth were abraded, but several posterior-most teeth of the right dentary were untouched exposing their buccal sides. Also, one left dentary tooth exposed an undamaged lingual surface (Fig 1f).

We followed the molding procedures of Kubo and Kubo [17]. Teeth of ZMNH M8961 that preserved shiny surfaces with none or minor visible damages were selected and cleaned by cotton swabs soaked in acetone. After the cleaning, molds were taken using high-resolution A-silicone dental impression material (Dr. Silicon regular type, DentKist, Inc., Korea). Buccal surfaces of six right dentary teeth that include two wear facets and one possible wear facet (Fig 1c) and one lingual surface of the left dentary tooth (Fig 1f) were molded.

The presence of microwear was observed using a confocal laser microscope (VK-9700, Keyence, Osaka, Japan) and when the preservation of the entire visual field was satisfactory (e.g. the existence of scratches and absence of large crack), the dental impressions were scanned using a violet laser (wavelength 408nm) with 10× and long-distance 100× objective lenses (numerical apertures 0.30 and 0.95 respectively). For a long-distance 100× objective lens, the scan pitches were 0.138 μm for one pixel in x- and y- axes, and for a 10× objective lens that are 1.379 μm. A vertical resolution was 1 nm for both lenses. By scanning, the light intensity of reflected laser, color information, and height data (Z position) were obtained for each XY coordinate point at each focus point. Using these data, VK-9700 generates real color ultra-depth images (photosimulation), from which dental microwear were visually checked. Each photosimulation was 1024×768 pixels, therefore covering the area of 141×106 μm with long-distance 100× objective lens and 1412×1059 μm with 10× objective lens.

Previous work has shown that scratch orientation differs apicobasally within the same wear facet in some ankylosaurs, reflecting a multiphasic, complex jaw mechanism [12]. Therefore, we scanned wear facets in apicobasal direction using 10× objective lens, to make the image of the entire apicobasal section by combining photosimulation images. To show apicobasal changes of orientation of scratches, one wear facet was divided into five apicobasal sections and another wear facet was divided into apical and basal halves. Then orientation of scratches observed in photosimularion of each section were measured using software imageJ [18]. Rose diagrams were generated from these data (see S1 File) for each section using the 'circular' package of a statistical software R [19, 20].

## Results

The teeth are preserved in both sides of the maxillae and dentaries of ZMNH M8961. However, the teeth in both maxillae are fragmentary, with buccal crown impressions of several left maxillary teeth being preserved in the matrix. The teeth are well-preserved in the posterior portion of the right dentary, with eight posterior adjoining teeth exposing their buccal view (Fig 1). A few teeth that are positioned more apical (dorsal) to adjacent teeth were probably functional when the animal lived and exhibited steeply inclined wear facets on their buccal sides (Fig 1b). Also, one tooth of the left dentary is well-preserved, exposing its lingual view. The dentary teeth are relatively small, with 6.4–6.7 mm mesiodistally, similar to those of other ankylosaurids, but smaller than those of nodosaurids [13]. Compared with other ankylosaurids, the teeth are slightly larger than those of *Pinacosaurus granger* (holotype AMNH 6523: 4 mm, ZPAL Mg D-II/1: 4–5 mm) [13, 21], and *Euoplocephalus* (AMNH 5405: 3–5 mm) [13], But smaller than those of *Gobisaurus* (9.5 mm) [22], and *Saichania* (PIN 3142/250: 7–8 mm) [13]. The crowns

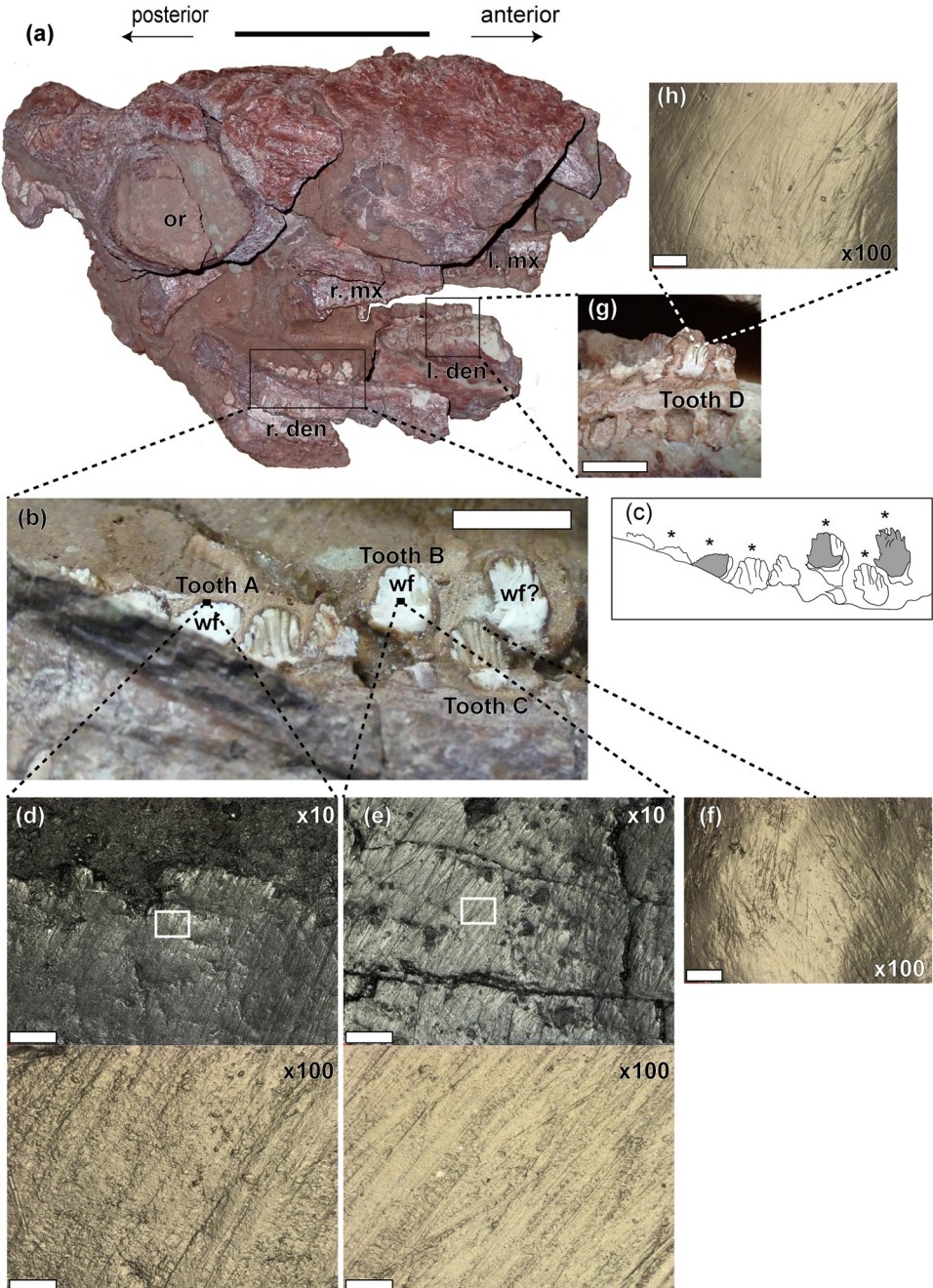

**Fig 1. Dental microwear of *Jinyunepelta*.** (a) Right rostrolateral view of the skull of *Jinyunpelta* ZMNH M8961. (b) close up view of buccal sides of the right posterior dentary tooth row and (c) its line drawing, in which wear facets were drawn in gray and asterisks indicate molded teeth. (d) Dental microwear at the apical border of wear facet on the tooth A. The white rectangle in 10× photosimulation indicates the area where 100× photosimulation was taken. (e) Dental microwear of the wear facet on the tooth B. The white rectangle in 10× photosimulation indicates the area where 100× photosimulation was taken. (f) Dental microwear of the buccal surface of the tooth C. (g) Close up of the lingual side of the left dentary showing the tooth D (h) Dental microwear of the lingual surface of the tooth D. All photosimularions were taken from molds using a confocal laser microscope (VK-9700) with either a 100× or 10× objective lens. The field of view is 140×105 μm for 100× lens and 1403×1052 μm for 10× lens. Photosimularions were mirrored in the horizontal direction to match its direction with the real tooth surfaces. Scale bars are 10 cm for (a), 1cm for (b) and (g), 0.2 mm for 10× photosimulation images and 20 μm for 100× photosimulation images. Abbreviations: den, dentary; l, left; mx, maxilla; or, orbit; r, right; wf, wear facet.

are buccolingually compressed, and leaf-shaped in buccal or lingual view, as in other ankylosaurs [23]. The dentary teeth of ZMNH M8961 bear 8–9 marginal denticles, similar to those of 'Tianzhenosaurus' (assigned to *Saichania* in Arbour and Currie [1]) [24], and *Kunbarrasaurus* (7–9 denticles) [25, 26], outnumber those of 'Crichtonsaurus bohlini' (IVPP V12745: 5–6 denticles) [27], juvenile *Pinacosaurus granger* (7 denticles) [28, 29], and *Liaoningosaurus* (7 denticles) [30, 31], but less than those of the holotype *Pinacosaurus granger* (AMNH 5405: 11 denticles). This denticles count of ZMNH M8961 is relatively low among ankylosaurs, which generally have 8–17 per tooth [23, 32, 33]. The buccal and lingual sides of crown is marked by longitudinal ridges extending ventrally from the denticles. The primary ridge positions almost in the center of the crown. The cingulum at the base of the crown is swollen, as in other ankylosaurs [23, 33]. Three highly worn teeth have very deep wear facet that occupies almost the entire buccal surface, similar to *Hungarosaurus* [13], and *Zuul* [33] and show clear contrast with the wear facets of *Saichania* (PIN 3142/250) that are more apically positioned [13], or the cases in juvenile *Pinacosaurus granger* [13, 28, 29], *Liaoningosaurus* [30, 31], and 'Crichtonsaurus bohlini' (IVPP V12745) [27], where little to no wear were observed on any of the teeth.

By examining photosimulated images of mold surfaces, it was confirmed that microwear was preserved in four teeth, which hereafter referred to as tooth A, B, C, and D (Fig 1b and 1h): buccal wear facets of two right dentary teeth (teeth A and B), the buccal surface of the right dentary tooth that does not have wear facet (tooth C), and the lingual surface of the left dentary tooth that do not have wear facet (tooth D). These tooth surfaces must have been affected by taphonomic processes to some extent as cracks were observed on the surface of the wear facet of the tooth B (Fig 1d). Taphonomic processes, however, tend to obliterate rather than create dental microwear [34], thus the observed orientation and density of scratches were unlikely to be affected by taphonomy.

The tooth that is positioned anterior to the tooth C (hereafter referred to as tooth E) also showed wear facet, however, its surface was likely chipped out postmortem as it did not show any dental microwear. Wear facets of tooth B and E extends almost the entire buccal side of a crown above the base of the cingulum. The basal portion of tooth A was embedded in the matrix and the entire exposed apical portion of its buccal surface formed a wear facet. Enamel-dentine interface on wear facets of *Jinyunpelta* was not clear as there is no visible step. Probably their enamel is very thin, in the photosimularion at the border of wear-facet (Fig 1c), the band of slightly shiny area was observed that is less than 0.1 mm thick, which probably is the layer of polished enamel.

The density of microwear features was low in surfaces without visible wear (tooth C and D: Fig 1f and 1h) compared with that in wear facets (Fig 1d and 1e), where the entire area was covered by numerous scratches. Microwear features on surfaces of tooth C and D were only scratches (Fig 1f and 1h). That of wear facets were mostly scratches but on the basal side of wear facets, there are some pit-like structures observed (Figs 2 and 3). The dental microwear of wear facet was probably produced by both attrition (tooth-tooth contact) and abrasion (tooth-food contact) whereas that of other surfaces was made solely by abrasion (tooth-food contact). Therefore, these difference between microwear features of wear facets and other surfaces meets the expectation that occlusal surface is much highly scratched than non-occlusal surface [35] and further confirming that observed scratches were not extensively affected by the postmortem taphonomic process.

Directions of scratches were more aligned on wear facets (Fig 1d and the lower half of Fig 1c) compared with that on surfaces without visible wear (tooth C and D: Fig 1e and 1f). The orientation of scratches changes between apicobasal positions within wear facets of teeth A and B. This orientation change can be observed clearer in the wear facet of tooth B compared with that of tooth A, because of its better preservation throughout the apicobasal axis (Fig 2).

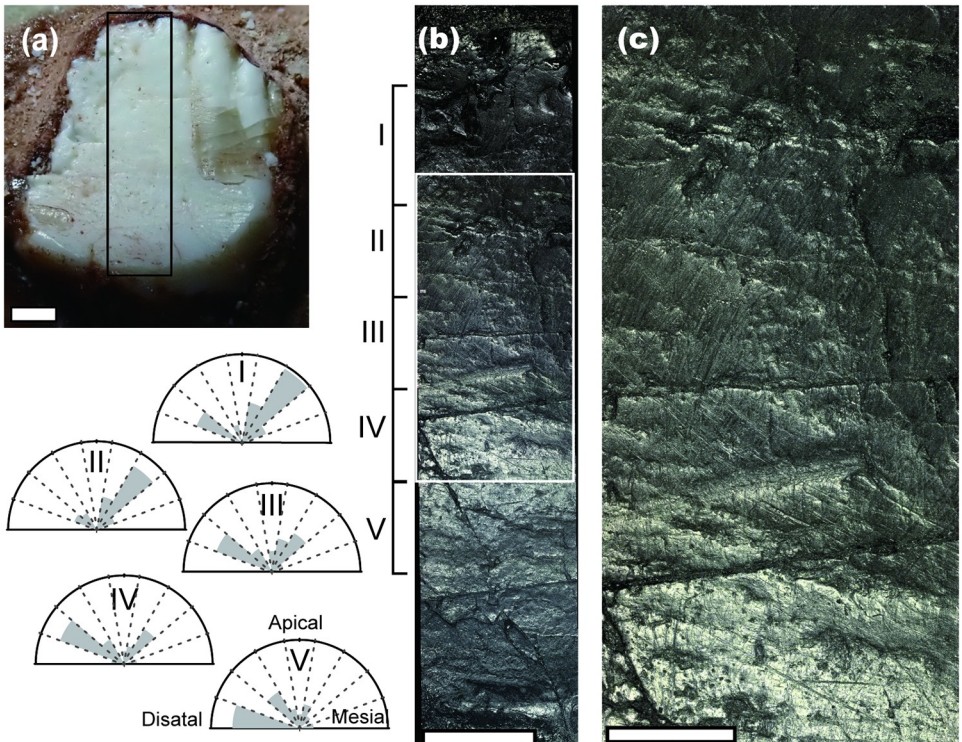

**Fig 2. Changes in the orientation of scratches along the apicobasal axis of wear facet of posterior right dentary tooth B.** (a) The buccal wear facet of the tooth B. (b) Dental microwear along the entire apicobasal axis of the wear facet that is enclosed by the black rectangle in Fig 2a. Scratch orientations were measured from five apicobasal sections (I-V) and rose diagrams were drawn to show orientations of scratches. Each section within a rose diagram represents 20 degrees and the length of each bar reflects the number of scratches oriented in that direction. Anatomical directions were shown in the rose diagram of section V. Numbers of measured scratches were 37, 42, 56, 49, and 40 for the section I to V respectively. (c) The close up of the middle region of the wear facet, where scratch orientations change apicobasally and is enclosed by the white rectangle in Fig 2b. Scale bars are 1 mm for (a) and (b) and 0.5mm for (c).

In the wear facet of tooth B, at the apical region, scratches were oriented apicomesial-distobasally about 40 degrees from the apicobasal axis (Fig 2). In the middle region, scratches of different orientations crossed each other; the ones that inclined mesially (anteriorly) about 40 degrees from the apicobasal axis and another one that inclined distally (posteriorly) about 60 degrees from the apicobasal axis. At the basal regions, Most scratches were almost horizontal, inclined distally (posteriorly) about 80 degrees from the apicobasal axis (Fig 2). In the wear facet of tooth A, scratches were inclined mesially (anteriorly) about 20 degrees from the apicobasal axis at the apical area of the wear facet (Figs 1c and 3). At the basal half of the wear facet, scratches of two orientations coexist, one that inclined mesially (anteriorly) about 30 degrees from the apicobasal axis that are located mostly in the mesial half and another one that inclined distally (posteriorly) about 60 degrees from the apicobasal axis that are located in the distal half (Fig 3).

## Discussion and conclusions

Scratch orientation found in wear facets of *Jinyunpelta* teeth changed from mesially inclined ones at the apical position of wear facet to more distally inclined ones at the basal position, which is almost identical to that of a nodosaurid ankylosaur *Hungarosaurus* [12: Fig 6]. Therefore, the chewing jaw movement of these two taxa must be similar. The orientation of

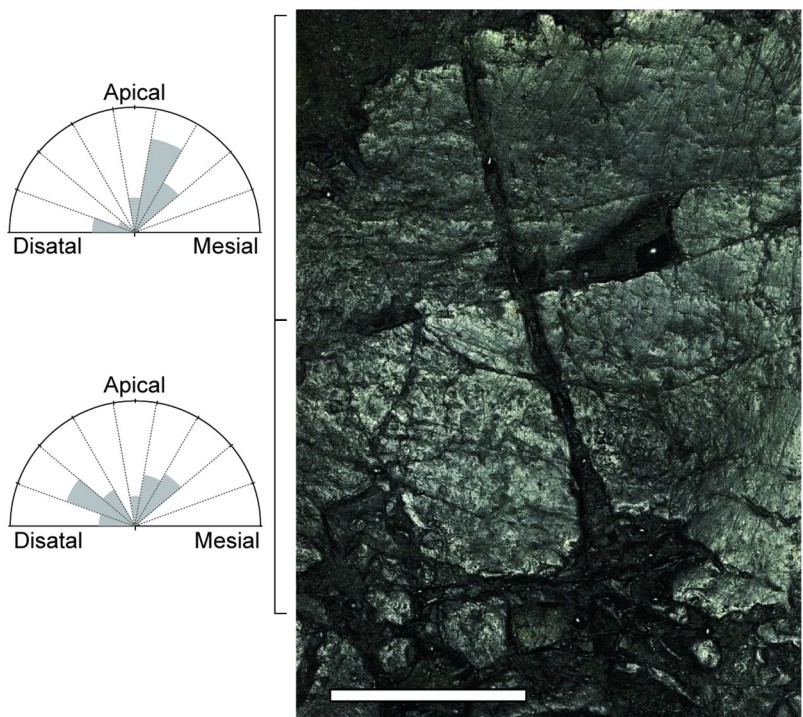

**Fig 3. Changes in the orientation of scratches along the apicobasal axis of wear facet of posterior right dentary tooth A.** Photosimulation of the buccal wear facet of tooth A taken by a 10× lens. Scratch orientations were measured from apical and basal sections and rose diagrams were drawn to show orientations of scratches. Numbers of measured scratches were 35 for the apical section and 52 for the basal section. For the explanation of a rose diagram see the caption of Fig 2. Scale bar represents 1mm.

scratches, which inclines 20 to 40 degrees mesially from the apicobasal axis in the apical region of the wear facet indicates that occlusion starts with orthal and possibly slight proal movement in *Jinyunpelta*. The existence of two scratch orientations that cross each other at the middle region of the wear facet might reflect an abrupt shift of lower jaw movement from orthal to the backward and slightly upward direction. The scarcity of low-angled scratches in the apical region of dentary teeth indicates, during this backward traction, the pressure on the apical region of wear facet from the antagonist upper tooth was not as strong as that on the basal region of the wear facet. The dominance of the low-angled, mesiobasal-apicodistal oriented scratches at the basal region of the wear facet indicates that the chewing cycle ended with the orthopalinal (sensu Varriale [36]) movement of the lower jaw.

As stated by Mallon and Anderson [3] "The microwear signal in one area of a tooth often differed from that in another area of the same tooth". In some dinosaurian taxa, the area within the wear facet that occluded with the antagonist tooth changes with the phase of chewing cycle. If the direction of jaw movements changes along with the phase of the chewing cycle (e.g. *Leptoceratops* and ankylosaurs with biphasal jaw movement), the orientation of dental microwear differs in different areas of wear facet [12, 36]. Therefore, preferably, observation of the entire apicobasal axis within a wear facet is needed to reconstruct the jaw movement of the entire chewing cycle. However, this is practically impossible in many cases as the preservation of the whole wear facet is rarely satisfactory to analyze dental microwear.

Scratches of *Jinyunpelta* are dense and fills the entire field of view (Fig 1d and 1e). Taphonomic processes tend to obliterate microwear features [34, 35], therefore these dense and

aligned scratches are probably produced when the animal was alive. Although only qualitative comparison is possible, microwear were compared between *Jinyunpelta* and other ankylosaurs using figures published in previous studies of Ősi [12, 13] and Mallon and Anderson [3]. It should be noted that different imaging devices and magnifications were used for different studies, so the results of this comparison should be taken with caution. Scratches were scarcer in *Hungarosaurus* [12: Fig 5, 13: Fig 11A], *Edmontonia* [13: Fig 11F and 11H], *Ankylosaurs* [13: Fig 12D], and *Gargoyleosaurus* [13: Fig 12H] than *Jinyunpelta*. Pits were more dominant in *Edmontonia* [13: Fig 11E and 11G], anterior maxillary tooth of *Euoplocephalus* (AMNH5405) [13: Fig 12E], *Saichania* [13: Fig 12C], and *Gargoyleosaurus* [13: Fig 12H]

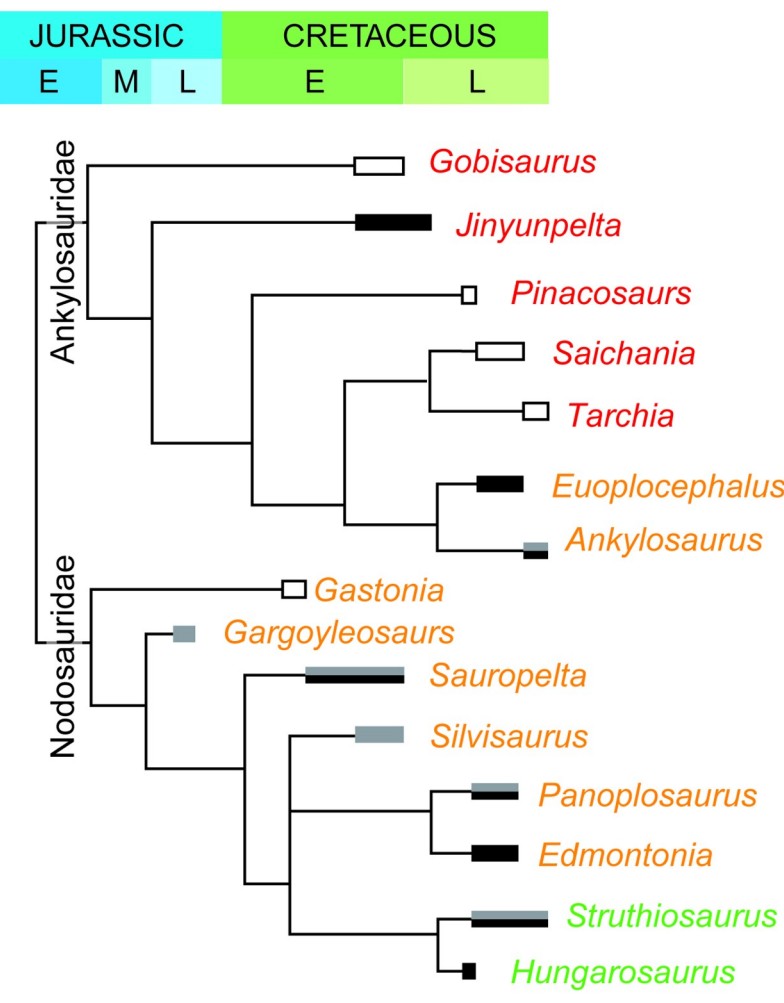

**Fig 4. Evolution of chewing mechanism in ankylosaurs.** The color of each genus name represents the continent they lived: red, Asia; orange, North America; green, Europe. Modified from Ősi et al. [13: Fig 15]. The phylogenetic relationship of ankylosaurs followed Zheng et al. [16].

compared with *Jinyunpelta*. Microwear of an isolated ankylosaurid tooth (TMP 1991.050.0014) [3: Fig 1] and maxillary tooth of *Euoplocephalus* (AMNH 5405) [13: Fig 12F] were dominated by dense and roughly aligned scratches that resemble the microwear of *Jinyunpelta*. These similarities and differences of dental microwear of *Jinyunpelta* and other ankylosaurs likely correspond to similarities and differences in their diets and jaw mechanism.

Among ankylosaurs, precise tooth-tooth occlusion was more widely seen among nodosaurids compared with ankylosaurids. Among ankylosaurids, the precise tooth occlusion was previously known only for the Late Cretaceous North American taxa (Fig 4) and the definite evidence of palinal jaw movement was known only for *Euoplocephalus* [11, 13]. The precise tooth-tooth occlusion with palinal jaw movement of *Jinyunpelta* confirmed in this study, however, indicates a sophisticated chewing jaw movement evolved much earlier among ankylosaurids, in the late Early (Albian) or early Late (Cenomanian) Cretaceous (Fig 4). It should be noted that, however, this conclusion is based on two teeth from one individual, therefore more studies on mid-Cretaceous ankylosaurids are awaited.

This study is the first to show solid evidence of precise tooth-tooth occlusion in an Asian ankylosaur. Asian ankylosaurs had been known to exhibit none or small nearly horizontal apical wear facets [13]. Therefore, steep wear facets of dentary teeth of *Jinyunpelta* that cover the most area of buccal surface is unique among Asian ankylosaurs. The lack of sophisticated feeding adaptations in Asian ankylosaurs and its presence in Late Cretaceous ankylosaurs of Europe and North America, was suspected to have been reflecting habitat differences among different continents [13]. The finding that contemporaneous Asian ankylosaurs, *Jinyunpelta* and *Gobisaurus*, differ in their chewing systems, indicated that these differences reflect local habitat differences and/or dietary differences among closely related taxa. The finding of tooth occlusion and biphasal jaw mechanism in *Jinyunpelta* that is phylogenetically nested within Asian taxa without tooth occlusion indicates the evolution of a complex feeding mechanism likely occurred convergently twice among ankylosaurids and at least once among nodosaurids, which suggest feeding mechanism of ankylosaurs was highly plastic. Roughly contemporaneous emergence of palinal jaw movement within both ankylosaurids and nodosaurids during the late Early to the early Late Cretaceous (Fig 4) corroborates the idea that these functional novelties of ankylosaurs jaw movement were triggered by a global phenomenon. The emergence and radiation of angiosperms may be a candidate for such a global change [13], although future studies on the timing of changes in paleoflora in different localities and diet reconstructions of associated megaherbivores are needed to test this hypothesis.

## Supporting information

**S1 File. Orientations of scratches in apicobasal sections of wear facet of tooth A and B.** (XLSX)

## Acknowledgments

We thank the following people for participating fieldwork in 2013: S. Gu (ZMNH), C. Chen (Jinyun Museum), K. Miyata (Fukui Prefectural Dinosaur Museum), and several local farmers. We thank C. Yu, A. Liu, and Y. Sheng (ZMNH) for preparing the specimens and H. Sakaki (University of Tokyo) for help in molding dental impressions at ZMNH. We also thank the editorial work of A. R. Fiorillo (Perot Museum of Nature and Science) and thorough reviews of two reviewers, A. Ősi (Hungarian Academy of Sciences) and F. J. Varriale (King's college) that significantly improved the manuscript.

## Author Contributions

**Conceptualization:** Tai Kubo.

**Data curation:** Tai Kubo, Wenjie Zheng, Xingsheng Jin.

**Formal analysis:** Mugino O. Kubo.

**Investigation:** Tai Kubo.

**Methodology:** Tai Kubo, Mugino O. Kubo.

**Project administration:** Xingsheng Jin.

**Resources:** Mugino O. Kubo.

**Visualization:** Tai Kubo.

**Writing – original draft:** Tai Kubo, Mugino O. Kubo.

**Writing – review & editing:** Wenjie Zheng, Xingsheng Jin.

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
