## [Decision Letter · Decision Letter 0]

14 Jan 2021

PONE-D-20-40068

Dental microwear of a basal ankylosaurine dinosaur, Jinyunpelta and its implication on evolution of chewing mechanism in ankylosaurs.

PLOS ONE

Dear Dr. Kubo,

Thank you for submitting your manuscript to PLOS ONE. After careful consideration, we feel that it has merit but does not fully meet PLOS ONE’s publication criteria as it currently stands. Therefore, we invite you to submit a revised version of the manuscript that addresses the points raised during the review process.

We look forward to receiving your revised manuscript.

Kind regards,

Anthony R Fiorillo

Academic Editor

PLOS ONE

Additional Editor Comments :

Both reviewers are of the opinion that this paper is a worthy contribution after revision. Each reviewer has provided very specific guidance on the manuscript itself, while also suggesting the use of English can use some polish. I encourage the authors to follow the reviewers' comments closely and that they resubmit to the journal at the appropriate time.

Journal Requirements:

Reviewers' comments:

Reviewer's Responses to Questions

**Comments to the Author**

1. Is the manuscript technically sound, and do the data support the conclusions?

Reviewer #1: Yes

Reviewer #2: Partly

2. Has the statistical analysis been performed appropriately and rigorously? 

Reviewer #1: N/A

Reviewer #2: N/A

3. Have the authors made all data underlying the findings in their manuscript fully available?

Reviewer #1: Yes

Reviewer #2: Yes

4. Is the manuscript presented in an intelligible fashion and written in standard English?

Reviewer #1: Yes

Reviewer #2: No

5. Review Comments to the Author

Reviewer #1: This is an important work for a better understanding of ankylosaur feeding mechanism, and the results, presented here, widens our knowledge on the feeding characters of Mesozoic herbivores in general. I would be more than happy to see this manuscript published but before doing this, I strongly recommend to complete the work with the followings:

Main comments:

- The descriptive part of the Results section should be more detailed. The original Zheng et al. (2018) paper did not details the dentition since teeth in the holotype specimen cannot be observed. Based on this new specimen, I suggest to add a separated block with the description of the few individual teeth in the preserved section, and their comparision with some other Asian ankylosaurs (e.g. the contemporaneous Gobisaurus) would be very useful. What about with the dentary tooth anterior to tooth C? It seems that at least one toth is also preserved in the maxilla. Some words about it (preservation, relative size, worn, not worn, etc.) would be also useful.

- Nothing is written about the general macrowear patterns of the teeth. How is the enamel preserved on the teeth? Is there any information about the enamel-dentine interface (where is it flush or step)? How is the relative wear ratio between the individual dentary teeth (i.e. A, B, C)? All this information would help a lot to better understand the jaw mechanism of Jinyunpelta and that of ankylosaurids.

- Authors used 4 teeth in this work that have microwear features preserved, but microwear features have been described in the Results section only from tooth B and A. I understand that on the other teeth (C, D) scratches are much less, but still present, according to MS line 126. Perhaps some basic comparison of these features between the individual teeth and between the teeth of Jinyunpelta and those of other ankylosaurs (see some data in Mallon and Anderson 2014, Ősi et al. 2016) can be added, e.g. pit-scratch number and ratio, main scratch orientation. The free Microware softwer is very easy to use for this purpose. A rose diagram might be also added to simply numerically demonstrate the orientation of scratches from the different regions of the teeth. Authors state that on the teeth where only abrasive wear is present, microwear features are much less than on the teeth bearing wear due to occlusion. What kind of other differences can be observed between the microwear features of the two type of wear?

- The authors write that they used a confocal microscope for getting high resolution pictures on the details of the microwear features. Why do they not get and analyzed then 3D data of these images since in this technice, as written in the Material and Methods section, a vertical component („height data (Z position)”) is also measured? The main point of a confocal microscope is that a 3D model can be got from a surface texture and the software can generate various data (complexity, anisotropy, etc. see Ungar 2003, Winkler et al. 2017) for comparative purposes. In this form, the images taken by the confocal microscope do not yield more than well-prepared scanning images.

- A basic drawing on the preserved tooth rows and the position and extension of their wear facets would be very informative. Perhaps Fig. 2B could be a bit larger to see as much details from the tooth crowns as possible. Unfortunately, the pdf I’ve got for review contained very poor resolution figures…

Some small comments, typos and corrections have been added directly in the annotated pdf.

Reviewer #2: General Comments

The manuscript represents the results of important original research examining facet development and dental microwear in a specimen of the ankylosaurid Jinyunepelta. This study is a valuable contribution to the literature on jaw mechanics in Ankylosauria as it fills a gap in our understanding of the paleoecology, biogeography, and evolution of mastication in Asian members of this group. It is a natural successor to the work of Ősi and others 2017 and builds on the discoveries and conclusions of that publication. The work is worthy of publication. I find the methods sound but the manuscript requires revision in the text and supplied figures.

An important aspect of this manuscript is the presentation of evidence for biphasal motion from microwear in Jinyunepelta. However, microwear from only a single tooth of one individual convincingly displays striations that indicate both orthal and palinal jaw action. The authors mention evidence of additional palinal wear on their “Tooth A” from the same specimen but do not provide a figure that highlights this wear. Figure 1 does show wear from tooth A but most of the striations are tilted mesially and it is difficult to discern scratches that indicate the palinal motion mentioned in the text. The authors should address how they know the wear on tooth B is not an artifact of preservation by figuring the wear they describe is present on tooth A. The preservation of real wear on tooth B is partially corroborated by its similarity to previously published ankylosaurs; however, the authors should also give serious consideration to amending the text of their conclusions to recognize the tenuous nature of having microwear from just one specimen. Further details regarding suggestions for how they can strengthen their evidence are included in the more specific comments by line below (Lines 140-142 & 149-152 and Lines 192-197).

There are grammatical and typographical errors in the manuscript that would benefit from further editing during the revision process. I have made some editorial suggestions by line in my review, but they are not exhaustive because my primary goal was the scientific merit of the study.

Some of the text would be better served if it were moved to other sections of the manuscript (ex. methods, conclusions). I have made note of these suggestions for reorganization in the specific comments by line under “Text of Manuscript”. (Lines 71-83, Lines 83-85, Line 86, Lines 87-89)

Text of Manuscript

Lines 29-32 & Lines 213-216 “Parallel evolution of the biphasal jaw mechanism, which contemporaneously occurred among two lineages of ankylosaurs, ankylosaurids and nodosaurids, might reflect changes in paleoflora during the late Early to the early Late Cretaceous”. There is unfortunately little evidence to support this, and the authors should consider amending the language of the introduction and conclusion to suggest that this is only a possibility and not a certainty. There could be alternative factors that caused the change. Ősi et al. 2017 suggest this and make the more general uncertain statement in their conclusions that this could be a possibility. See their bullet point #6 on pages 565-566, “One possible reason for the appearance of these functional morphological novelties might be paleofloral change during the Cretaceous, but this cannot be supported at the moment.” The reason this cannot be supported is because it is beyond the scope of the research that is in these papers. Correlation and timing in the turnover of both the dinosaurian fauna and the paleoflora would be needed, and even then, that would only be circumstantial correlation and not direct evidence of relationship.

Line 42: Perhaps change “Among two main ankylosaurian” to “Between the two main ankylosaurian” Among is used for comparing objects greater than two.

Lines 45-46 “Not only diets but also jaw mechanism were implicated in ankylosaur dental microwear” I understand the intent of this sentence, but it could be stated more clearly, Try “In addition to diet, jaw mechanics are revealed via ankylosaur dental microwear”.

Lines 46-47 “Contrary to the expectation that ankylosaurs adopted simple orthal jaw movement due to their small and simple leaf-shaped teeth.” Who is expecting this and why? Please provide a reference for this expectation. Provide justification via referral to previously published literature for this statement other than the shape and size of their teeth. There is nothing inherent about small teeth that necessitates orthal mastication. However, simple leaf shaped teeth have previously been interpreted to indicate orthal mastication. Please provide a reference for this statement. There are previous authors who have suggested that Ankylosaurs chewed through orthal mastication (Orthal pulping). References in reference #10 should help. Also see Weishampel & Jianu, 2000 and Weishampel and Normal, 1989. Additional text may not be necessary, only referral to publications that claimed orthal mastication.

Lines 55-56 “Within these groups with tooth occlusion, palinal jaw movement evolved convergently several times.” This sentence seems redundant as it says basically the same thing as the previous sentence. Perhaps remove it to be more concise. Or if the authors intended further convergence within each of these groups perhaps connecting it with the previous sentence by saying “and within each of these groups tooth occlusion and palinal movement appears to have evolved several times.”

Lines 56-57: “An Early Cretaceous nodosaurid Sauropelta might have been adopted palinal jaw movements” Remove ‘been’ from this sentence.

Line 57: reverse the order of these words “be also”

Lines 66-68. This is a cumbersome sentence; try rewording to something like; “The dental microwear of more ankylosaur specimens is needed to determine if the many convergences found buy Ősi and others 2017 are the true pattern of evolution or an artifact of sample size.”

Line 70: Change “were demanded” to “are needed”

Lines 71-83: The text on these lines describes the specimen and the condition of the teeth. They therefore belong more appropriately at the beginning of the materials and methods section as a description of the material examined in this analysis.

Line 79: Change “Lately” to “Recently”

Line 86: Was it the intention of the authors to indicate that Asian forms rarely showed high angle and not low-angle? In their paper Ősi and others 2017 state: “In Asian ankylosaurids (e.g. Gobisaurus, Pinacosaurus spp., Saichania, Tarchia) tooth wear is either restricted to the apical cusps slightly exposing the underlying dentine, or it is more extensive basally as a smooth surface, yet does not penetrate the thin enamel. Steep wear facets, similar to those seen in nodosaurids, are present neither on lingual/labial sides of the crown, nor on the cingulum.”. They also note that Gobisaurus and Tarchia have low angle facets.

Lines 83-85: “Further, a few teeth of the right dentary that positioned more apical (dorsal) to adjacent teeth, which were probably functional when the animal lived, exhibited steeply inclined wear facets on their buccal sides (Fig 1b).” This statement is more appropriately placed in the results section.

Line 86: “This contradicts with the notion of Ősi et al. [10] that Asian ankylosaurids rarely showed low-angled apical wear, implying the absence of precise tooth occlusion.” This sentence might be better reserved for the discussion and conclusions section as it is follows from what microwear and facet shape look like on Jinyunpelta.

Lines 87-89: “In this study, therefore, we observed the dental microwear of Jinyunpelta to deduce its jaw movement and to reconstruct the evolution of the feeding mechanism in Asian and Cretaceous ankylosaurids.” Once lines 71-85 have been moved to the sections suggested above, lines 87-89 might be better placed after line 70. This line is a natural continuation of the ideas the authors have been setting up in the introduction and the purpose of their work. Try removing the word “therefore” from the sentence to make it concise.

Line 107: There appear to be 8 visible dentary teeth in Fig 1b, perhaps placing a marker or asterisk above or below the teeth in the figure to indicate which of the 6 were molded.

Line 109: Change “Preservation of dental impressions” to “The presence of microwear”

Line 110” How did the authors determine what satisfactory preservation is? Please provide a small description of what that means and what criteria was used to determine if preservation was satisfactory?

Line 121: Perhaps altering the beginning of this sentence will make it more concise. “Previous work has shown that scratch orientation differs apicobasally…”

Line 127: Perhaps change (Fig. 1) to (Fig. 1b,f) because those are the figures that show the identification of the teeth examined.

Line 134: Perhaps indicating this in the text by telling the reader with referral to figures. For instance, this can be easily accomplished with only a slight change to line 134 to “The density of scratches was low in non-wear facets (Fig. 1e,g) and high in wear facets (Fig 1c,d).

Line 136-137: “Therefore, this difference in microwear densities meets the expectation” What expectation are being referred to here? This can be as simple as quoting and adding a reference for this. If it is the same as (reference 15) then the authors could reiterate that reference for this sentence.

Line 139: Perhaps change (Fig. 1) to be more specific like the suggestion for line 134 above.

Lines 140-142 & 149-152: “This orientation change can be observed clearer in the wear facet of tooth B compared with that of tooth A, because of its better preservation throughout the apicobasal axis (Fig 2).” “At the basal half of the wear facet, preservation of microwear was not as good as the apical half, but it seemed scratches of two orientations coexist, one that inclined mesially (anteriorly) about 30 degrees from the apicobasal axis and another one that inclined distally (posteriorly) about 60 degrees from the apicobasal axis” The authors should include in Fig 2 whatever images they have of photosimulations that depict microwear on the basal wear facet of tooth A showing (posterior) palinal motion. Despite the poor preservation of tooth A, wear from this tooth is important evidence that is missing from this publication. Currently only one tooth is figured (Tooth B) with clear indications of biphasal motion. The reader is required to trust that Tooth A also contains scratches that indicate palinal motion. Therefor the notion that biphasal jaw action is present in Jinyunepelta is based on one tooth from one specimen. That is tenuous evidence and would be more convincing if the authors depicted any other teeth from this specimen that supports palinal motion. If the claim is made that there are two sets of scratches visible on tooth A then images can be provided of what is seen that support the statement, no matter how poor the preservation might be.

Line 162: Perhaps remove “less” to be more concise.

Line 163: “facet and more inclined” change to “facet to more distally inclined”

Line 167: “occlusion start with orthal movement” or it may mean slight proall + orthal motion

Line 169: “backward and slightly upward direction” This type of motion has been referred to as orthopalinal by several authors Varriale, (2016), Mallon and Anderson (2014) & Nabavizadeh (2020)

• Nabavizadeh A. New reconstruction of cranial musculature in ornithischian dinosaurs: implications for feeding mechanisms and buccal anatomy. The Anatomical Record. 2020;303: 347–362. doi:10.1002/ar.23988

Line 183-184 “These taxa, including Jinyunpelta, also resemble in their chewing manner that adopted biphasal jaw movement.” This sentence is difficult to understand, try rephrasing it to be clearer.

Line 184-185 “Scratches of Jinyunpelta are much denser than other ankylosaurs that fills the entire field of view and are almost countless” Can the authors provide any quantitative defense of this statement? For example, number per unit area. Alternatively, they could qualitatively discuss why they think this is real by direct comparison with figures from published results of other ankylosaurs and why it is not an artifact of preservation or taphonomy. The sentence is also difficult to understand try using another word than countless, perhaps “too numerous” would be a better choice.

Lines 192-197: As detailed above, only one tooth from one specimen shows evidence of biphasal motion. This is not enough evidence for solid confirmation of biphasal motion. Perhaps provide more evidence of figures from tooth A that help solidify the claims of the manuscript and/or amend the language of the text to be more uncertain by suggesting that microwear from more individuals of Jinyunpelta are necessary to confirm this result.

Figures and Figure Captions

Figure 1:

• Text and abbreviations in figure are small and difficult to read. Please consider making the text larger or bolded for visibility. Some text does not stand out well against the photographs. Perhaps increasing the font size and bolding the letters will help. The authors could also try adding a white shadow behind the black text to make it stand out.

• The text above many of the scale bars is illegible because it does not stand out against the background of the photograph. Please make the text larger and bolder or place the scale bar dimensions in the figure captions for each sub-letter.

• (a) does not seem like an exactly right lateral view of the specimen but one that is offset somewhat to be a rostrolateral.

• Perhaps increasing the size of the dots in the lines or changing them to dashes to make them more visible.

• Add rectangles to (c) and (d) to show the location of the 100x photosimulations in the 10x images.

Figure 2:

• Placing a scale bar in a) would be helpful. It would be instructive to have scale bars in (b) and (c) as well

• In figure 2 a) is bolded but b) and c) are not. Bold them so that these sub-figure labels have greater visibility and to be consistent among them.

Figure 3:

• The authors may have misinterpreted the double-colored bars in figure 3 compared to what is depicted in a similar figure in Ősi et al. 2017. In Ősi et al. 2017 orthal (dorsal) movement of the dentary is green and palinal (posterior) movement is blue. When they are present together this indicates biphasal because orthal is one phase and palinal is the other phase (See Ősi et al. 2017, page 563, under jaw mechanisms read their section 3). Figure 3 of this manuscript is showing grey as orthal and black as biphasal which is redundant because one of the phases of biphasal motion is orthal. Perhaps change grey and black to be orthal and palinal or color the entire bar of these ankylosaurs to be black for biphasal.

• Placing the labels Ankylosauridae and Nodosauridae to the side of their respective branches rather than over the branches would make these labels more readable.

References

Some of the references have all major words capitalized but others do not. Perhaps review the references list to standardize citations to a common formatting.

6. PLOS authors have the option to publish the peer review history of their article (what does this mean?). If published, this will include your full peer review and any attached files.

Reviewer #1: **Yes: **Attila Ősi

Reviewer #2: **Yes: **Frank J. Varriale

---

## [Author Response · Author response to Decision Letter 0]

15 Feb 2021

Reviewer #1: This is an important work for a better understanding of ankylosaur feeding mechanism, and the results, presented here, widens our knowledge on the feeding characters of Mesozoic herbivores in general. I would be more than happy to see this manuscript published but before doing this, I strongly recommend to complete the work with the followings:

Main comments:

- The descriptive part of the Results section should be more detailed. The original Zheng et al. (2018) paper did not details the dentition since teeth in the holotype specimen cannot be observed. Based on this new specimen, I suggest to add a separated block with the description of the few individual teeth in the preserved section, and their comparison with some other Asian ankylosaurs (e.g. the contemporaneous Gobisaurus) would be very useful. What about with the dentary tooth anterior to tooth C? It seems that at least one tooth is also preserved in the maxilla. Some words about it (preservation, relative size, worn, not worn, etc.) would be also useful.

We wrote a new section that describes Jinyunpelta‘s dentitions at the beginning of the Results section. We compared tooth size and denticle number with that of other ankylosaurids (lines 142-161). The tooth anterior to the tooth C show wear facet but it does not preserve microwear, probably it surface was chipped out postmortem, we added an explanation about this tooth (lines 170-171). The teeth in both maxillae are fragmentary, with only buccal crown impressions of several left maxillary teeth being preserved in the matrix (lines 136-138), thus we could not obtain enough information for maxillary tooth.

- Nothing is written about the general macrowear patterns of the teeth. How is the enamel

preserved on the teeth? Is there any information about the enamel-dentine interface (where is it ﬂush or step)? How is the relative wear ratio between the individual dentary teeth (i.e. A, B, C)? All this information would help a lot to better understand the jaw mechanism of Jinyunpelta and that of ankylosaurids.

We wrote sentences in the Results section to describe macrowear and enamel-dentine interface of Jinyunpelta as follows (lines 170-177). “The tooth that is positioned anterior to the tooth C (hereafter referred to as tooth E) also showed wear facet, however, its surface was likely chipped out postmortem as it did not show any dental microwear. Wear facets of tooth B and E extends almost the entire buccal side of a crown above the base of the cingulum. The basal portion of tooth A was embedded in the matrix and the entire exposed apical portion of its buccal surface formed a wear facet. Enamel-dentine interface on wear facets of Jinyunpelta was not clear as there is no visible step. Probably their enamel is very thin, in the photosimularion at the border of wear-facet (Fig. 1c), the band of slightly shiny area was observed that is less than 0.1 mm thick, which probably is the layer of polished enamel”

- Authors used 4 teeth in this work that have microwear features preserved, but microwear

features have been described in the Results section only from tooth B and A. I understand that on the other teeth (C, D) scratches are much less, but still present, according to MS line 126. Perhaps some basic comparison of these features between the individual teeth and between the teeth of Jinyunpelta and those of other ankylosaurs (see some data in Mallon and Anderson 2014, Ősi et al. 2016) can be added, e.g. pit-scratch number and ratio, main scratch orientation. The free Microware software are very easy to use for this purpose. A rose diagram might be also added to simply numerically demonstrate the orientation of scratches from the diﬀerent regions of the teeth. Authors state that on the teeth where only abrasive wear is present, microwear features are much less than on the teeth bearing wear due to occlusion. What kind of other diﬀerences can be observed between the microwear features of the two type of wear?

We described and compare microware feature Jinyunpelta and other ankylosaurs (lines 245-259). Microwears of wear facets and other surfaces were compared in more detail (lines 178-182). We also made rose diagrams for tooth A and B to visually show how scratch orientations change along the vertical axis (Fig. 2 and 3). 

By using the word “non-wear facet”, we may have misunderstood as if these surfaces have visible macrowear due to abrasion. Although at the microscopic level these surfaces have scratches due probably to abrasion, we used “non-wear facet” just to indicate the surface outside of wear facet or the surface of a tooth without wear facet. To avoid this misunderstanding, we reworded all “non-wear facet” in the manuscript.

- The authors write that they used a confocal microscope for getting high resolution pictures on the details of the microwear features. Why do they not get and analyzed then 3D data of these images since in this technice, as written in the Material and Methods section, a vertical

component („height data (Z position)”) is also measured? The main point of a confocal microscope is that a 3D model can be got from a surface texture and the software can generate various data (complexity, anisotropy, etc. see Ungar 2003, Winkler et al. 2017) for comparative purposes. In this form, the images taken by the confocal microscope do not yield more than well-prepared scanning images. 

We aimed to apply DMTA analysis to Jinyunpelta when scanned by a confocal microwear. However, we found DMTA analysis needs extremely good preservation. When a 10x lens was used for scanning, there are lots of noises on 3D images, which made it impossible to analyze. Using a 100x lens, we can get a good 3D image, but it requires superb preservation of the fossil that is not the case for our specimen (image of x100 lens is equivalent in its pixel resolution to about x700 image of SEM). However, we found photosimulation images we got were good enough to conduct conventional 2D dental microwear analysis and write this manuscript. We would love to look for well-preserved specimens of ankylosaurs and conduct DMTA analysis in the future.

- A basic drawing on the preserved tooth rows and the position and extension of their wear facets would be very informative. Perhaps Fig. 2B could be a bit larger to see as much details from the tooth crowns as possible. Unfortunately, the pdf I’ve got for review contained very poor resolution ﬁgures…

We enlarged Fig. 2b and also added a line drawing of Fig. 2b as Fig. 3c.

Some small comments, typos and corrections have been added directly in the annotated pdf.

 We checked all corrections on the PDF files and addressed all of them. In the following, we describe our revisions, which are not simple typo corrections.

perhaps some basic data should be added, i.e. it is an ankylosaur, form which age and locality.

 The sentence was added “Jinyunpelta sinensis is a basal ankylosaurine dinosaur excavated from the mid Cretaceous Liangtoutang Formation of Jinyun County, Zhejiang Province, China ” (lines 20-21).

and in the European Hungarosaurus as well, see Ősi et al. 2014, 2016.

 The sentence was rewritten from “Among ankylosaurids, the biphasal jaw movement was previously only known from the Late Cretaceous North American taxa,” to “The biphasal jaw movement was widely observed among nodosaurids, among ankylosaurids, it was previously only known from the Late Cretaceous North American taxa, and not known among Asian ankylosaurids.” (lines 28-30).

Would be good to specify it better with definitive ages based on fossils

 Rewritten from “adaptations emerged among ankylosaurids much earlier than previously thought.” to “much earlier (during Albian or Cenomanian) than previously thought (during Campanian)”. (lines 31-32)

Is the exact number, even the minimum number of individuals unknown? Or there is five associated/articulated specimens, plus isolated material?

 Rewritten from “more than five individuals” to “five associated specimens　and isolated materials of Jinyunpelta” (lines 82-83).

I do not think that the results, presented here, contradict with the notion of Ősi et al., just simply shows that these type of wear facets are rare but present in some asian ankylosaurs.

 This sentence was deleted. 

these pixel size is not the same as written in lines

 These are pixel numbers for images we obtained from our confocal microscope. It may be changed during figure making, uploading, and so on.

This "non-wear facet" phrase is controversial to me suggesting that there were no wear at all. However, this is not the case, since even if there was no occlusion of the upper and lower teeth, there should have been at least tooth-food contact.

So, I may suggest to use simply "wear facet" for this, or differentiate them as "attritive wear facet" and "abrasive wear facet".

 We did not find visible wear facet or visible wear on these surfaces. Scratches were observed only by microscope. As suggested “non-wear facet” is misleading so we avoided using this term throughout the manuscript.

During attrition both tooth-tooth and tooth-food contacts play a role since both is related to food processing.

 Rewritten from “by attrition (tooth-tooth contact)” to “by both attrition (tooth-tooth contact) and abrasion (tooth-food contact)”. (lines 182-183)

Please add here, e.g. compare Fig. 1d and Fig.1e

 The sentence was rewritten from “. Directions of scratches were more aligned on wear facets compared with non-wear facets (Fig1).” to “Directions of scratches were more aligned on wear facets (Fig. 1d and the lower half of Fig. 1c) compared with that on surfaces without visible wear (tooth C and D: Fig. 1e, f).” (lines 188-189)

I think these scratchere are not really inclined, but simply oriented. So, I suggest to add a precize direction of the scratches, in this case: scratches are oriented apicomesial-distobasally about 30 degrees.

 Rewritten as “scratches were oriented apicomesial-distobasally about 40 degrees from the apicobasal axis”. (line 193)

But why? This type of movement has allowed contact between the apical region of one tooth and the basal part of the occluding tooth, resulting in the low-angled scratches at the base (eroded cingular region) of the wear facet.

We assumed “During this backward traction, the apical region of wear facets did not have contact with the antagonist tooth.” Because we could not find many horizontal scratches on the apical region of dentary teeth. However, as the reviewer pointed out apical region must have contacted with the basal region of the antagonist tooth. So we rewrite it as “The scarcity of low-angled scratches in the apical region of dentary teeth indicates, during this backward traction, the pressure on the apical region of wear facet from the antagonist upper tooth was not as strong as that on the basal region of the wear facet. ” (lines 230-233).

Please, rephrase.

 We rewrite “the evolution of a complex feeding mechanism could be evolved easily among ankylosaurs.” to “the evolution of a complex feeding mechanism likely occurred convergently twice among ankylosaurids and at least once among nodosaurids, which suggest feeding mechanism of ankylosaurs was highly plastic”. (lines 282-284)

Please, indicate (perhaps on Fig part C and D) from which part the wear facet are these 100x images.

 The change was made in Fig. 1 as suggested.

Scale bar for these figure parts would be very useful.

 Scale bars were added to Fig. 2.

Reviewer #2: General Comments

The manuscript represents the results of important original research examining facet development and dental microwear in a specimen of the ankylosaurid Jinyunepelta. This study is a valuable contribution to the literature on jaw mechanics in Ankylosauria as it fills a gap in our understanding of the paleoecology, biogeography, and evolution of mastication in Asian members of this group. It is a natural successor to the work of Ősi and others 2017 and builds on the discoveries and conclusions of that publication. The work is worthy of publication. I find the methods sound but the manuscript requires revision in the text and supplied figures.

An important aspect of this manuscript is the presentation of evidence for biphasal motion frommicrowear in Jinyunepelta. However, microwear from only a single tooth of one individual convincingly displays striations that indicate both orthal and palinal jaw action. The authors

mention evidence of additional palinal wear on their “Tooth A” from the same specimen but do not provide a figure that highlights this wear. Figure 1 does show wear from tooth A but most of the striations are tilted mesially and it is difficult to discern scratches that indicate the palinal motion mentioned in the text. The authors should address how they know the wear on tooth B is not an artifact of preservation by figuring the wear they describe is present on tooth A. The preservation of real wear on tooth B is partially corroborated by its similarity to previously published ankylosaurs; however, the authors should also give serious consideration to amending the text of their conclusions to recognize the tenuous nature of having microwear from just one specimen. Further details regarding suggestions for how they can strengthen their evidence are included in the more specific comments by line below (Lines 140-142 & 149-152 and Lines 192-197).

 We added the figure of tooth A as figure 3 and apicobasal change of scratch orientation was analyzed for both tooth A and B. We also added caution that the conclusion of this study was drawn only from two teeth of one specimen and future studies on this topic are awaited. (lines 266-267). 

There are grammatical and typographical errors in the manuscript that would benefit from further editing during the revision process. I have made some editorial suggestions by line in my review, but they are not exhaustive because my primary goal was the scientific merit of the study.

 We corresponded to all suggestions given, please see below 

Some of the text would be better served if it were moved to other sections of the manuscript (ex. methods, conclusions). I have made note of these suggestions for reorganization in the specific comments by line under “Text of Manuscript”. (Lines 71-83, Lines 83-85, Line 86, Lines 87-89) 

 We moved all sections as suggested, please see below.

Text of Manuscript

Lines 29-32 & Lines 213-216 “Parallel evolution of the biphasal jaw mechanism, which contemporaneously occurred among two lineages of ankylosaurs, ankylosaurids and nodosaurids, might reflect changes in paleoflora during the late Early to the early Late Cretaceous”. There is unfortunately little evidence to support this, and the authors should consider amending the language of the introduction and conclusion to suggest that this is only a possibility and not a certainty. There could be alternative factors that caused the change. Ősi et al. 2017 suggest this and make the more general uncertain statement in their conclusions that this could be a possibility. See their bullet point #6 on pages 565-566, “One possible reason for the appearance of these functional morphological novelties might be paleofloral change during the Cretaceous, but this cannot be supported at the moment.” The reason this cannot be supported is because it is beyond the scope of the research that is in these papers. Correlation and timing in the turnover of both the dinosaurian fauna and the paleoflora would be needed, and even then, that would only be circumstantial correlation and not direct evidence of relationship.

 We have deleted this sentence from the abstract and rewrite the last part of the manuscript to express that more studies are needed to prove the coevolution of paleoflora and megaherbivore dinosaurs as “Roughly contemporaneous emergence of palinal jaw movement within both ankylosaurids and nodosaurids during the late Early to the early Late Cretaceous (Fig 4) corroborates the idea that these functional novelties of ankylosaurs jaw movement were triggered by a global phenomenon. The emergence and radiation of angiosperms may be a candidate for such a global change [13], although future studies on the timing of changes in paleoflora in different localities and diet reconstructions of associated megaherbivores are needed to test this hypothesis..” (lines 284-290)

Line 42: Perhaps change “Among two main ankylosaurian” to “Between the two main

ankylosaurian” Among is used for comparing objects greater than two.

 Change was made following the suggestion (line 44).

Lines 45-46 “Not only diets but also jaw mechanism were implicated in ankylosaur dental　microwear” I understand the intent of this sentence, but it could be stated more clearly, Try “In addition to diet, jaw mechanics are revealed via ankylosaur dental microwear”.

 This sentence was rewritten following the suggestion (line 47-48).

Lines 46-47 “Contrary to the expectation that ankylosaurs adopted simple orthal jaw movement due to their small and simple leaf-shaped teeth.” Who is expecting this and why? Please provide a reference for this expectation. Provide justification via referral to previously published literature for this statement other than the shape and size of their teeth. There is nothing inherent about small teeth that necessitates orthal mastication. However, simple leaf shaped teeth have previously been interpreted to indicate orthal mastication. Please provide a reference for this statement. There are previous authors who have suggested that Ankylosaurs chewed through orthal mastication (Orthal pulping). References in reference #10 should help. Also see Weishampel & Jianu, 2000 and Weishampel and Normal, 1989. Additional text may not be necessary, only referral to publications that claimed orthal mastication.

 Following the suggestion, Weishampel & Jianu (2000), Weishampel and Norman (1989), and Hwang (2005) were cited (line 49).

Lines 55-56 “Within these groups with tooth occlusion, palinal jaw movement evolved

convergently several times.” This sentence seems redundant as it says basically the same thing as the previous sentence. Perhaps remove it to be more concise. Or if the authors intended further convergence within each of these groups perhaps connecting it with the previous sentence by saying “and within each of these groups tooth occlusion and palinal movement appears to have evolved several times.”

 This sentence was removed as suggested.

Lines 56-57: “An Early Cretaceous nodosaurid Sauropelta might have been adopted palinal jaw movements” Remove ‘been’ from this sentence.

Line 57: reverse the order of these words “be also”

 Changes were made following the suggestions (line 57 and 58).

Lines 66-68. This is a cumbersome sentence; try rewording to something like; “The dental

microwear of more ankylosaur specimens is needed to determine if the many convergences found buy Ősi and others 2017 are the true pattern of evolution or an artifact of sample size.”

 This sentence was rewritten as suggested (lines 66-68).

Line 70: Change “were demanded” to “are needed”

 This word was changed as suggested (line 70).

Lines 71-83: The text on these lines describes the specimen and the condition of the teeth. They therefore belong more appropriately at the beginning of the materials and methods section as a description of the material examined in this analysis.

 These lines were moved to the materials and methods section (line 75-92).

Line 79: Change “Lately” to “Recently”

 Changed as suggested (line 87).

Line 86: Was it the intention of the authors to indicate that Asian forms rarely showed high angle and not low-angle? In their paper Ősi and others 2017 state: “In Asian ankylosaurids (e.g. Gobisaurus, Pinacosaurus spp., Saichania, Tarchia) tooth wear is either restricted to the apical cusps slightly exposing the underlying dentine, or it is more extensive basally as a smooth surface, yet does not penetrate the thin enamel. Steep wear facets, similar to those seen in nodosaurids, are present neither on lingual/labial sides of the crown, nor on the cingulum.”. They also note that Gobisaurus and Tarchia have low angle facets. 

 Yes, we miswrote “high” as “low”. This sentence was deleted to correspond to other comments.

Lines 83-85: “Further, a few teeth of the right dentary that positioned more apical (dorsal) to

adjacent teeth, which were probably functional when the animal lived, exhibited steeply inclined wear facets on their buccal sides (Fig 1b).” This statement is more appropriately placed in the results section.

 This sentence was moved to the result section (lines 139-141).

Line 86: “This contradicts with the notion of Ősi et al. [10] that Asian ankylosaurids rarely showed low-angled apical wear, implying the absence of precise tooth occlusion.” This sentence might be better reserved for the discussion and conclusions section as it is follows from what microwear and facet shape look like on Jinyunpelta.

 Thank you for the suggestion. Adding this sentence to the conclusion section may be redundant because in the conclusion section there is a sentence “Therefore, steep wear facets of dentary teeth of Jinyunpelta that cover the most area of the buccal surface is unique among Asian ankylosaurs”. So, we simply removed this sentence.

Lines 87-89: “In this study, therefore, we observed the dental microwear of Jinyunpelta to deduce its jaw movement and to reconstruct the evolution of the feeding mechanism in Asian and Cretaceous ankylosaurids.” Once lines 71-85 have been moved to the sections suggested above, lines 87-89 might be better placed after line 70. This line is a natural continuation of the ideas the authors have been settng up in the introduction and the purpose of their work. Try removing the word “therefore” from the sentence to make it concise.

 We followed this suggestion, moved sentences and deleted “therefore” (lines 70-72).

Line 107: There appear to be 8 visible dentary teeth in Fig 1b, perhaps placing a marker or asterisk above or below the teeth in the figure to indicate which of the 6 were molded.

 Asterisks were added above the teeth that were molded in Fig. 1c that is newly added line drawing of Fig. 1b, and were mentioned in the caption of Fig. 1 (line 96).

Line 109: Change “Preservation of dental impressions” to “The presence of microwear”

 Changed as suggested (line 115).

Line 110” How did the authors determine what satisfactory preservation is? Please provide a small description of what that means and what criteria was used to determine if preservation was satisfactory?

The explanation “(e.g. the existence of scratches and absence of large crack)” was added (lines 116-117).

Line 121: Perhaps altering the beginning of this sentence will make it more concise. “Previous work has shown that scratch orientation differs apicobasally…”

Line 127: Perhaps change (Fig. 1) to (Fig. 1b,f) because those are the figures that show the

identification of the teeth examined.

Changes were made as suggested (lined 126 and 163).

Line 134: Perhaps indicating this in the text by telling the reader with referral to figures. For

instance, this can be easily accomplished with only a slight change to line 134 to “The density of scratches was low in non-wear facets (Fig. 1e,g) and high in wear facets (Fig 1c,d).

Line 136-137: “Therefore, this difference in microwear densities meets the expectation” What

expectation are being referred to here? This can be as simple as quoting and adding a reference for this. If it is the same as (reference 15) then the authors could reiterate that reference for this sentence.

 I added words “meets the expectation that occlusal surface is much highly scratched than non-occlusal surface” and added the citation Teaford (2007) that visualized the difference between dental microwear of non-occlusal and occlusal surfaces of mammals and how it is lost due to taphonomic processes (lines 185-187). 

Line 139: Perhaps change (Fig. 1) to be more specific like the suggestion for line 134 above.

 Changes were made accordingly (lines 188-189). 

Lines 140-142 & 149-152: “This orientation change can be observed clearer in the wear facet of tooth B compared with that of tooth A, because of its better preservation throughout the apicobasal axis (Fig 2).” “At the basal half of the wear facet, preservation of microwear was not as good as the apical half, but it seemed scratches of two orientations coexist, one that inclined mesially (anteriorly) about 30 degrees from the apicobasal axis and another one that inclined distally (posteriorly) about 60 degrees from the apicobasal axis” The authors should include in Fig 2 whatever images they have of photosimulations that depict microwear on the basal wear facet of tooth A showing (posterior) palinal motion. Despite the poor preservation of tooth A, wear from this tooth is important evidence that is missing from this publication. Currently only one tooth is figured (Tooth B) with clear indications of biphasal motion. The reader is required to trust that Tooth A also contains scratches that indicate palinal motion. Therefor the notion that biphasal jaw action is present in Jinyunepelta is based on one tooth from one specimen. That is tenuous evidence and would be more convincing if the authors depicted any other teeth from this specimen that supports palinal motion. If the claim is made that there are two sets of scratches visible on tooth A then images can be provided of what is seen that support the statement, no matter how poor the preservation might be.

 We added a new figure (Fig. 3) to show the scratch orientation of tooth A and analyzed the scratch orientation of the upper and lower half to show its change along apicobasal orientation.

Line 162: Perhaps remove “less” to be more concise.

Line 163: “facet and more inclined” change to “facet to more distally inclined”

Line 167: “occlusion start with orthal movement” or it may mean slight proall + orthal motion

 Changes were made according to suggestions (lines 223, 224, and 228).

Line 169: “backward and slightly upward direction” This type of motion has been referred to as orthopalinal by several authors Varriale, (2016), Mallon and Anderson (2014) & Nabavizadeh (2020)

 Rewritten as “… the chewing cycle ended with the orthopalinal (sensu Varriale [36]) movement of lower jaw” (lines 234-235).

Line 183-184 “These taxa, including Jinyunpelta, also resemble in their chewing manner that

adopted biphasal jaw movement.” This sentence is difficult to understand, try rephrasing it to be clearer.

 I rewrote this section to make a more detailed comparison between the microwear of Jinyunpelta and other ankylosaurs. Consequently, this sentence was deleted. 

Line 184-185 “Scratches of Jinyunpelta are much denser than other ankylosaurs that fills the entire field of view and are almost countless” Can the authors provide any quantitative defense of this statement? For example, number per unit area. Alternatively, they could qualitatively discuss why they think this is real by direct comparison with figures from published results of other ankylosaurs and why it is not an artifact of preservation or taphonomy. The sentence is also difficult to understand try using another word than countless, perhaps “too numerous” would be a better choice.

 It is hard to conduct quantitative comparisons between Jinyunpelta and other ankylosaurids, for this revision as we do not have molds of other ankylosaurids and comparisons between images taken by a different type of microscope (optical, scanning electron, and confocal laser) may not be appropriate. Therefore, we visually compared microwear images of Jinyunpelta and other ankylosaurs qualitatively and added a section in lines 245-259.

Lines 192-197: As detailed above, only one tooth from one specimen shows evidence of biphasal motion. This is not enough evidence for solid confirmation of biphasal motion. Perhaps provide more evidence of figures from tooth A that help solidify the claims of the manuscript and/or amend the language of the text to be more uncertain by suggesting that microwear from more individuals of Jinyunpelta are necessary to confirm this result.

 As written above, we added the figure and analysis of tooth A, also we stated the caution as follows “It should be noted, however, that this conclusion is based on two teeth from one individual, therefore more studies on mid-Cretaceous ankylosaurids are awaited.” (lines 266-267)

Figures and Figure Captions

Figure 1:

• Text and abbreviations in figure are small and difficult to read. Please consider making the text larger or bolded for visibility. Some text does not stand out well against the photographs. Perhaps increasing the font size and bolding the letters will help. The authors could also try adding a white shadow behind the black text to make it stand out.

• The text above many of the scale bars is illegible because it does not stand out against the

background of the photograph. Please make the text larger and bolder or place the scale bar

dimensions in the figure captions for each sub-letter.

• (a) does not seem like an exactly right lateral view of the specimen but one that is offset

somewhat to be a rostrolateral.

• Perhaps increasing the size of the dots in the lines or changing them to dashes to make them more visible.

• Add rectangles to (c) and (d) to show the location of the 100x photosimulations in the 10x

images.

Texts were enlarged and changed to bold font. White rims were added to black texts and black rims were added to white texts. Scale bars were heightened and black rims were added. Scale bar dimensions were removed from the figure and mentioned in the caption. Dot lines were enlarged and rectangles were added to (c) and (d) to indicate areas of 100x photosimulations. The caption was changed from “right lateral” to “right rostrolateral” in line 90. 

Figure 2:

• Placing a scale bar in a) would be helpful. It would be instructive to have scale bars in (b) and (c) as well

• In figure 2 a) is bolded but b) and c) are not. Bold them so that these sub-figure labels have

greater visibility and to be consistent among them.

 Scale bars were added to Fig. 2b and c and within figure texts were bolded.

Figure 3:

• The authors may have misinterpreted the double-colored bars in figure 3 compared to what is depicted in a similar figure in Ősi et al. 2017. In Ősi et al. 2017 orthal (dorsal) movement of the dentary is green and palinal (posterior) movement is blue. When they are present together this indicates biphasal because orthal is one phase and palinal is the other phase (See Ősi et al. 2017, page 563, under jaw mechanisms read their section 3). Figure 3 of this manuscript is showing grey as orthal and black as biphasal which is redundant because one of the phases of biphasal motion is orthal. Perhaps change grey and black to be orthal and palinal or color the entire bar of these ankylosaurs to be black for biphasal.

• Placing the labels Ankylosauridae and Nodosauridae to the side of their respective branches rather than over the branches would make these labels more readable.

 In the legend of figure 3, “biphasal” was rewritten as “palinal” and the placement and direction of “Ankylosauridae” and “Nodosauridae” were changed following the suggestions.

References

Some of the references have all major words capitalized but others do not. Perhaps review the references list to standardize citations to a common formating.

 All citations were checked to standardize the format.

---

## [Editor Report · Decision Letter 1]

17 Feb 2021

Dental microwear of a basal ankylosaurine dinosaur, Jinyunpelta and its implication on evolution of chewing mechanism in ankylosaurs.

PONE-D-20-40068R1

Dear Dr. Kubo,

We’re pleased to inform you that your manuscript has been judged scientifically suitable for publication and will be formally accepted for publication once it meets all outstanding technical requirements.

Kind regards,

Anthony R Fiorillo

Academic Editor

PLOS ONE
---

## [Editor Report · Acceptance letter]

19 Feb 2021

PONE-D-20-40068R1 

Dental microwear of a basal ankylosaurine dinosaur, *Jinyunpelta* and its implication on evolution of chewing mechanism in ankylosaurs. 

Dear Dr. Kubo:

I'm pleased to inform you that your manuscript has been deemed suitable for publication in PLOS ONE. Congratulations! Your manuscript is now with our production department. 

Kind regards, 

on behalf of

Dr. Anthony R Fiorillo 

Academic Editor

PLOS ONE